# Changes in the Urine Metabolomic Profile in Patients Recovering from Severe COVID-19

**DOI:** 10.3390/metabo13030364

**Published:** 2023-02-28

**Authors:** Robert Rosolanka, Peter Liptak, Eva Baranovicova, Anna Bobcakova, Robert Vysehradsky, Martin Duricek, Andrea Kapinova, Dana Dvorska, Zuzana Dankova, Katarina Simekova, Jan Lehotsky, Erika Halasova, Peter Banovcin

**Affiliations:** 1Clinic of Infectology and Travel Medicine, Jessenius Faculty of Medicine in Martin (JFM CU), University Hospital in Martin, Comenius University in Bratislava, 036 01 Martin, Slovakia; 2Clinic of Internal Medicine-Gastroenterology, Jessenius Faculty of Medicine in Martin (JFM CU), University Hospital in Martin, Comenius University in Bratislava, 036 01 Martin, Slovakia; 3Biomedical Centre BioMed, Jessenius Faculty of Medicine in Martin, Comenius University in Bratislava, Mala Hora 4, 036 01 Martin, Slovakia; 4Clinic of Pneumology and Phthisiology, Jessenius Faculty of Medicine in Martin (JFM CU), University Hospital in Martin, Comenius University in Bratislava, 036 01 Martin, Slovakia; 5Department of Medical Biochemistry, Jessenius Faculty of Medicine in Martin (JFM CU), Comenius University in Bratislava, 036 01 Martin, Slovakia

**Keywords:** COVID-19, urine, metabolomics, ketosis, NMR, discrimination, random forest

## Abstract

Metabolomics is a relatively new research area that focuses mostly on the profiling of selected molecules and metabolites within the organism. A SARS-CoV-2 infection itself can lead to major disturbances in the metabolite profile of the infected individuals. The aim of this study was to analyze metabolomic changes in the urine of patients during the acute phase of COVID-19 and approximately one month after infection in the recovery period. We discuss the observed changes in relation to the alterations resulting from changes in the blood plasma metabolome, as described in our previous study. The metabolome analysis was performed using NMR spectroscopy from the urine of patients and controls. The urine samples were collected at three timepoints, namely upon hospital admission, during hospitalization, and after discharge from the hospital. The acute COVID-19 phase induced massive alterations in the metabolic composition of urine was linked with various changes taking place in the organism. Discriminatory analyses showed the feasibility of successful discrimination of COVID-19 patients from healthy controls based on urinary metabolite levels, with the highest significance assigned to citrate, Hippurate, and pyruvate. Our results show that the metabolomic changes persist one month after the acute phase and that the organism is not fully recovered.

## 1. Introduction

Severe acute respiratory syndrome corona virus 2 (SARS-CoV-2), which caused the recent COVID-19 pandemic, is still the subject of ongoing worldwide research in various fields. The SARS-CoV-2 virus primarily targets the lung tissue. However, its impact on other organ systems, such as the gastrointestinal system, circulatory, or central nervous system, has been well described [1,2,3,4,5]. The virus, present in the organism, was able to induce changes in the metabolism of various molecules (e.g., glucose, cholesterol, amino acids, tryptophan, etc.) that are not only important for the virus itself due to survival and reproduction, but also play a crucial role in immune response regulation. Such changes at the level of metabolic processes can be referred to as metabolic reprogramming [6,7]. Viruses are in general not metabolically active, however; in infected patients, they can cause changes at the level of metabolic processes in the human body. Influencing some of these host factors may result in a different disease outcome for the infected individual. Thus, correcting the activity of some metabolic processes can be one of the promising methods for controlling infection and enhancing metabolism [8].

Metabolomics is a relatively new research area that focuses mostly on the profiling of selected molecules and metabolites within the organism. Due to progressive development in the field of data analytics and bioinformatics, metabolomics has started to be widely applied in clinical and biomedical research. The analysis of the human metabolome can contribute to a better understanding of the onset and progression of various diseases and also proves to be a promising tool in the search for new diagnostic and prognostic markers of these diseases [9,10,11]. There is even emerging evidence of a strong correlation between such complex environments as the microbiome and the metabolomic profile of various experimental models [12,13,14]. Thus, understanding the human gut microbiome with the flow of metabolites across several trophic levels, and the creation of microbial biomas, may be the key factor for understanding its health impact [15]. Among other things, several studies that focused on 16S rRNA gene sequencing have shown that COVID-19 can alter not only the upper respiratory microbiome but also the gut microbiome with the loss of a large number of important microbial strains [16]. Thus, the ability to determine the precise metabolomic and metagenomic status of human microbiota could have substantial clinical implications in the future. Since the beginning of the pandemic, research on the use of metabolomics on COVID-19 has been greatly accelerated [17]. SARS-CoV-2 infection itself can lead to major disturbances in the metabolite profile of the infected individuals. These changes can occur at several levels and in various bodily fluids. Although most studies are focused on serum metabolite levels and longitudinal alterations, a few studies have focused on the impairment of urinary metabolism [18,19] especially considering the omics profiling during the disease recovery stage. There is increasing evidence that in some patients, postacute sequelae of SARS-CoV-2 infection may persist for a long time, with significant consequences for their further quality of life. Understanding the pathophysiological processes that take place individually not only in the acute phase of infectious diseases but also in the convalescence stage, appears to be an important milestone for the future and may ultimately benefit the patient. In this study, we wish to freely follow up on our previous work, where we analyzed metabolic changes in blood plasma at three timepoints, focusing on the patients who survived the severe course of SARS-CoV-2 infection requiring hospitalization and oxygen supplementation. Similar to the previous study, where we monitored metabolomic changes in blood plasma in the three timepoints—the acute phase at the time of hospital admission, one week later during hospitalization, and after one month in the recovery phase when the acute phase had passed [20]. In this study, we aimed to evaluate metabolomic changes in the urine. As the majority of the included patients were the same individuals, we discuss the observed changes in relation to current knowledge and to the alterations resulting from changes in blood plasma metabolome, as described previously [20]. In further describing the impact of the altered metabolome on the biochemical and physiological function of an organism, we run a discriminatory analysis to judge the discriminatory power of the system relative to healthy controls.

## 2. Materials and Methods

### 2.1. Patients

The sampling for the study was performed from November 2021 until February 2022. A total of twenty-four (24) patients were enrolled in the study with the following exclusion criteria: age under 18 years, pregnancy, and unwillingness or incapability to sign the informed consent. All included patients had compensated chronic diseases. All participants signed the informed consent. The study was approved by the Ethics Committee of the Jessenius Faculty of Medicine in Martin, Comenius University in Bratislava, Slovakia (Certification code at the US Office for Human Research Protection, US Department of Health and Human Services: IRB00005636 Jessenius Faculty of Medicine, Comenius University in Martin IRB # 1) with identification number: EK 65/2021. Every patient included in the study had a SARS-CoV-2 infection confirmed by a polymerase chain reaction (PCR). The study was focused on patients with typical SARS-CoV-2 infection symptoms (fever, cough, and dyspnea) with the goal of achieving a high level of homogeneity (in the means of the COVID-19 presentation) within the cohort. Therefore, only hospitalized patients with a severe course of COVID-19, requiring oxygen supplementation, though not invasive artificial pulmonary ventilation, (based on the National Institutes of Health/NIH/criteria) with X-ray or CT-confirmed pneumonia were included.

The first urine in the morning was sampled into a plastic tube. The first sample (sample A, *n* = 24) was taken within 24 h of admission to the hospital. The second sample (sample B, *n* = 22) was taken on days 5–8 (based on the course of the hospitalization, e.g., early discharge led to a shorter interval between samples A and B) and the third sample (sample C, *n* = 22) was taken at a median of 42 days (29–54 days) after the first sample (Figure 1). Blood for standard biochemical and hematological analysis was obtained during every sample-taking occasion. The groups of samples were labeled group A, group B, and group C based on the timepoint labeling. The characterization of the patient group is shown in Table 1 and patients’ biochemical and hematological results at the sampling times are listed in Table 2. 

### 2.2. Controls

As control samples, the first urine in the morning was used from 24 subjectively healthy volunteers aged at a median of 48 years (IQR 27); female/male, 15/9, with a negative antigen SARS-CoV-2 test on the sampling day; and reporting not having had acute COVID-19 disease or any positive SARS-CoV-2 test in the three months before sampling.

### 2.3. Urine Sampling

Urine was sampled into plastic tubes, centrifuged (2000 rpm, 4 °C, 20 min) within 2 h, and aliquots were stored at −80 °C. After thawing, the urine was centrifuged at 2000 rpm, at room temperature, for 20 min. For the measurement, 400 µL of centrifuged urine were carefully mixed with 200 µL of a stock solution consisting of (500 mM phosphate buffer, pH-meter reading 7.40, and 0.25 mM TSP-d_4_ (3-(trimethylsilyl)-propionic-2,2,3,3-d_4_ acid sodium salt in deuterated water) as a chemical shift reference in deuterated water). For measurement, 550 μL of the final mixture was transferred into a 5 mm NMR tube. 

### 2.4. NMR Data Acquisition

NMR data were acquired on a 600 MHz NMR spectrometer Avance III from Bruker equipped with a TCI (triple resonance) cryoprobe. Initial settings (water suppression frequency, pulse calibration, and shimming) were performed on an independent sample and adopted for measurements. The samples were stored in a Sample Jet at approx. 6 °C before measurement for a maximal time of 3 h. We modified standard Bruker profiling protocols as follows: profiling 1D NOESY with presaturation (noesygppr1d): FID size 64 k, dummy scans 4, number of scans 32, spectral width 20.4750 ppm; COSY with presaturation was acquired for 10 randomly chosen samples (cosygpprqf): FID size 4 k, dummy scans 8, number of scans 1, spectral width 16.0125 ppm; homonuclear *J*-resolved (jresgpprqf): FID size 8 k, dummy scans 16, number of scans 4; profiling CPMG with presaturation (cpmgpr1d, L4 = 126, d20 = 3 ms): FID size 64 k, dummy scans 4, number of scans 256, spectral width 20.0156 ppm. All experiments were conducted with a relaxation delay of 4 s, and all data were once zero filled. An exponential noise filter was used to introduce 0.3 Hz line broadening before the Fourier transform. Samples were measured at 310 K and randomly ordered for acquisition. The evaluation was performed on cpmg-acquired spectra.

### 2.5. Data Normalization

Preanalysis normalization is often based on biological parameters such as creatinine, specific gravity, and osmolality. The creatinine concentration is one of the most commonly used reference factors since the urinary creatinine level is supposed to reflect the overall concentration of metabolites [21]. However, although creatinine excretion is assumed to remain constant across individuals, many factors such as diet, muscle mass, age, and physical activity can affect creatinine levels [22]. The second reliable method for evaluating overall urine metabolite concentration is osmolality, which, together with specific gravity, completes the trio of the most important normalization methods. Osmolality and creatinine are in a direct relationship [22], so the results should be comparable after normalization. In this work, we decided to normalize the data to the urinary level of creatinine as the easiest, most reliable, and most feasible method. 

### 2.6. Data Analysis

A chemical shift of 0.000 ppm was assigned to the TSP-d_4_ signal. Spectra were solved using an internal metabolite database, an online human metabolome database (www.hmdb.ca accessed in 1 November 2022) [23], chenomx software (free trial version), and the literature [24,25,26]. For all compounds, the multiplicity of peaks was confirmed in J-resolved spectra, and homonuclear cross peaks were confirmed in a COSY spectra. Spectra were integrated manually and normalized to the integral of the creatinine peak at 3.05 ppm. Metabolites not having appropriate signals for the evaluations—peak overlap, or with unambiguous peak assignment—were excluded from further evaluation. Similarly, metabolites without well-resolved peaks in more than 3/4 of all samples were also left out of the statistical evaluation.

The null hypothesis of equality of population medians among groups was tested by the nonparametric Kruskal–Wallis test, with Dunn’s post hoc test for pairwise comparison. We applied Bonferroni correction to the data to avoid type I error, considering *p*-value 0.0083 as a threshold to claim significance. Principal component analysis (PCA) and the receiver operating characteristic curves (ROC) derived from the random forest (RF) algorithm were performed using MetaboAnalyst [27,28]. 

## 3. Results

Altogether, 21 metabolites were identified in the urine samples. However, signals from only 14 were appropriate for further evaluation, meeting the strict criteria, which were no peak overlap, unambiguous assignment, and the possibility of evaluation in a sufficient number of samples. In the first step, we employed PCA analysis to obtain an estimation of the group differences in the course of the COVID-19 disease, together with the controls (Figure 2A), and also in more detail a comparison of patients at the individual COVID-19 sampling times against the controls (Figure 2B–D). The relative concentrations of urine metabolites related to urine creatinine level were used as input variables. When comparing all three sampling times with the controls, samplings A and B differed from healthy individuals, whereas sampling in time C partially overlapped with controls. 

PCA analysis revealed the possibility of successful statistical separation of the patients in the sampling times A and B against controls. However, for the estimation of the real discriminatory power of the system, a different approach must be used. PCA analysis is often accompanied by PLS-DA, which is not only a 2D projection of multidimensional data but is also enriched with a discriminatory algorithm. However, PLS-DA often overfits data [29] and therewith may result in overoptimistic outcomes. To avoid this shortage, we employed a cross-validated random forest discriminatory algorithm that is not known to overfit the data and is also stable to outliers [30]. As input variables, we used the relative concentrations of metabolites in urine related to creatinine. The results from the RF are summarized in Table 3. As a quantitative parameter to judge the system performance, we used the area under the curve (AUC) value derived from the ROC curve (receiver operating characteristic curve). The system was able to discriminate patients in sampling point A almost ideally against control subjects with AUC = 1 and slightly weaker but still, excellent discrimination was attained for patients in sampling point B against the controls, with AUC = 0.99. Discrimination of patients one month after hospitalization against controls was not ideal, as was already expected from PCA analyses (Figure 2D). However, the resulting value AUC = 0.90 signaled a relatively successful discrimination. The results from further combinations in discriminatory analyses (A-B, A-C, B-C, A-B-C, A-B-C-ctrl) are included in Appendix A.

The statistical evaluation of the differences between the population medians marked 13 metabolites as significantly changed between the groups, with a *p*-value < 0.05. Bonferroni correction to avoid type I errors with a *p*-value of 0.0083 to claim significance, excluded TMA, DMA, and 1-methylniacinamid from significantly changed metabolites. The results, together with the relative changes derived from the medians, are summarized in Table 4.

## 4. Discussion

### 4.1. Metabolic Changes in Urine

Turnover from glycolytic to ketone-body metabolism in hospitalized COVID-19 patients was demonstrated in previous studies [20,31,32], and just the ability to normalize energy metabolism was suggested as one of the key parameters determining disease outcome [31]. Once in a ketotic state, the human body produces two fundamental substances, 3-hydroxybutyrate for energy utilization and acetoacetate for energy release, where the energy utilization according to need is ensured by their mutual conversion. As a side reaction, acetoacetate decarboxylates nonenzymatically to acetone, which, rather than being metabolized, is excreted from the body by urine, sweat, or breath. In the COVID-19 patients we observed, an increase in acetone level in urine in the acute COVID-19 phase (samplings A and B, Figure 3), as it was similar to our previous study where a ketotic-like state was detected during hospitalization [24]. As ketosis subsided during COVID-19 recovery [24], acetone levels in the urine of patients one month after hospitalization, decreased to being comparable to the level of control subjects. 

Ketone bodies, found to be increased in COVID-19 patients in the first week after hospitalization [20,31], are synthesized from acetyl-CoA produced by excessive beta-oxidation of fatty acids in the mitochondrion. Both processes, the transfer as well as subsequent oxidation, require the necessary presence of carnitine [33], which can be assumed with the diet or synthesized by the body. At the time of enhanced fatty acids catabolism, more fatty acids need to be transported into the hepatic mitochondria for oxidation, and the demand for carnitine in an organism increases. As shown in the work by Berry-Kravis et al., the levels of plasma carnitine increased in patients on a ketotic diet [34]. Carnitine in urine follows the trend of carnitine in plasma [35], where reabsorption by the kidney plays an important role. Tubular resorption in the kidney of free carnitine takes place at between 98% and 99% unless the transporters become saturated, and further carnitine overload leads to increased carnitine urine levels [36,37]. In parallel with acute ketosis found in hospitalized COVID-19 patients [24], overproduced carnitine is to be excreted by the urine. In our study, elevated carnitine levels in urine were more pronounced at sampling time B, where most likely the carnitine production exceeded the need. The urine carnitine level in COVID-19 patients decreased to the level of controls one month after hospitalization when the ketotic condition passed and the fatty acids utilization decreased by metabolism alterations.

Hippurate, a mammalian microbial cometabolite, is formed in the mitochondrial matrix in the liver and kidney (renal cortex) [38] and is a normal constituent of the endogenous urinary metabolite profile. Its relative urinary abundance seems to be linked with metabolic state, as it was reversely related to BMI [39], and increased in urine in type one diabetes [40] and type two patients [41]. Extensive investigations of the role of gut microbiota in the metabolism of polyphenolic compounds (hippurate precursors) [42] showed that that antibiotic-induced suppression of the gut microbiota results in a reduction in the excretion of hippurate and related metabolites [38,42]. According to the last finding, we supposed that, in this study, the observed decrease in urinary hippurate in COVID-19 patients was related to antibiotic treatment that the patients underwent. Interestingly, the urinary hippurate level did not fully recover one month after hospitalization, which suggests still insufficient intestinal microbiota colonization one month after acute COVID-19 disease. 

In our work we observed the interesting dynamics of urine hypoxanthine levels found to be increased in COVID-19 patients on the first day against the controls, followed by a more pronounced increase one week later and achieving the control level one month after hospitalization. Elevated hypoxanthine levels in blood serum were found in COVID-19 patients previously by Dogan et al. [43]. Inflammation and hypoxia induce the release of ATP from intracellular stores to extracellular space, and its conversion to adenosine monophosphate (AMP), which is then metabolized to adenosine, inosine, and hypoxanthine. This process makes hypoxanthine concentration in the blood a sensitive parameter of tissue hypoxia and ischemia [44]. After tissue damage following ATP depletion, there may be a prolonged excessive excretion that lasts at least two to three days [45]. Based on this, the observed increase in urinary hypoxanthine in COVID-19 patients could be explained as a result of hypoxic and inflammatory tissue damage. The most prominent increase observed one week after hospitalization (Figure 3), could be due to a particular time delay in hypoxanthine clearance, backwardly reflecting the pathological processes in the organism and disease severity. 

Citrate is a central metabolite of the energy-forming Krebs cycle. It is freely filtered at the kidney glomerulus, and then, in the amount of 65–90%, is reabsorbed in the proximal tubule, leaving about 10–35% of the filtered excreted in the urine [46,47]. Interestingly, it can be renally metabolized in the Krebs cycle to HCO_3_^-^ and consequently represents a potential base, indicating its possible, though not extensively researched role, in the acid–base balance [47]. Further, it also serves as metabolic fuel for the kidney and an endogenous inhibitor of calcium kidney stones [47], and its low urinary levels are a known risk factor for the development of calcium kidney stones [48]. Its urinary levels were markedly decreased in the initial COVID-19 phase, though systematically increased during hospitalization and one month after, heading towards the levels of the controls. The etiology of hypocitraturia is very diverse, including the ketotic state [42,43]. In connection with this, urinary citrate levels in COVID-19 patients inversely followed the dynamics of ketosis described in blood plasma in our previous paper [20] and could be linked to the stabilization of mitochondrial energy metabolism during the tendency to recover the energy metabolism from the utilization of ketone-bodies back to glycolysis [49]. 

The next metabolite showing significant changes in urine in COVID-19 patients was formate, which is produced from a variety of metabolic sources. Its principal function is as a source of one carbon groups included in purine synthesis and the provision of methyl groups for synthetic, regulatory, and epigenetic methylation reactions with the active folate [50]. Its elevated clearance in urine may be a sign of impaired one carbon metabolism.

A very similar time course of urinary alanine levels was observed in this study in urine, as it was observed in the blood plasma in COVID-19 patients; a decrease in the acute phase in time of hospital admission [20], achieving levels of healthy individuals already one week as well as one month later. Alanine (together with glutamine) is responsible for the detoxification of extrahepatic tissues and muscles from metabolically produced tissue-toxic NH_3_. Decreased alanine plasma [20], as well as urine levels found in this study, may suggest a slowdown in the nitrogen shuttle into the liver and potency for liver gluconeogenesis. In other words, alterations in the metabolic rate in an organism. This suggestion is supported by the corresponding increase of BCAAs and BCKAs in blood plasma in the previous study [20]. Furthermore, in plasma [20] as well as in urine (Table 4), decreased pyruvate signals a slowdown of glycolysis, which is in parallel with the observed hyperglycemia—a decreased utilization of glucose [20]. An increase in tyrosine may be produced by its overproduction from phenylalanine or insufficient utilization. Based on our previous study [20] we suggest that in patients with acute severe COVID-19, the use of tyrosine is lowered which may result in the underproduction of thyroid hormones and tyrosine-derived neurotransmitters such as dopamine and norepinephrine.

### 4.2. Multivariate and Discriminatory Analysis

The 2D visualization of metabolic features in urine in patients at the time of the acute COVID-19 phase and one month after hospitalization, together with healthy controls, is shown in Figure 2A. The urine metabolomes belonging to the greatest clinical COVID-19 manifestation, sampling points A and B, are visually similar, however, they are different from the metabolome in sampling point C and the controls, which are partially overlapping. This trend was confirmed in the next PCA analyses, where binary systems A, B, and C were evaluated against controls (Figure 2B–D). 

The high potential of metabolomics in the field of biomarkers was already demonstrated by the successful discrimination of COVID-19 patients in the acute phase against controls using blood plasma levels of metabolites [20,31]. In this work, we also employed the RF algorithm that includes cross validation via balanced subsampling. It works with two-thirds of the data for training and the rest for testing for regression, and about 70% of the data for training and the rest for testing during classification to overcome the negative features of training and testing on the same data. This approach may partially substitute the validation of an independent dataset; however, it cannot fully replace clinical validation. We used relative concentrations of metabolites in urine, expressed by the spectral integrals of particular NMR regions related to the signal of creatinine as input variables for the RF algorithm. In the case of highly correlating predictors, RF may label some of them as unimportant, therefore the RF was run ten times. Within the RF reruns, metabolites are permuted a little in the order of importance. 

The results, as summarized in Table 3, showed excellent discrimination of patients at sampling point A against controls with AUC = 1 (Figure 4), where the metabolites of the highest importance were hippurate, citrate, pyruvate, alanine, and hypoxanthine. For samples taken one week later, sampling point B, the discrimination was almost ideal with AUC = 0.99 (Figure 4), which was achieved using relative abundances of the metabolites citrate, hippurate, carnitine, hypoxanthine, and pyruvate. Slightly weaker with AUC = 0.90 (Figure 4), though still very good, was the discrimination of patients in the post-COVID phase one month after hospitalization, when all evaluated metabolites were included. These findings point out the extensive metabolomic changes in urine caused by severe COVID-19 course in hospitalized patients, present not only in the acute phase but also one month after. 

Note that when compared with our previous study [20], the urine metabolome performed comparably with the blood plasma metabolome with discriminating patients in sampling time A against the controls (both AUC = 1), and also similarly in sampling time B (AUC = 0.948 for blood plasma against 0.993 for urine) and sampling time C (AUC = 0.932 for blood plasma and AUC = 0.901 for urine). As both metabolomes are closely linked, this result is not surprising, and from the discriminatory point of view, both are biological samples of the same informative value. 

## 5. Conclusions

The longitudinal dynamics of alterations in urine metabolites in patients hospitalized with severe COVID-19 disease were monitored at three timepoints. The acute COVID-19 phase induced massive alterations in the metabolic composition of urine linked with various changes taking place in the organism. The increase in the urine levels of acetone and carnitine levels in the first week after hospital admission was linked with turnover in energy metabolism from glycolytic to ketone bodies, already known from other studies on blood plasma. Levels of both metabolites normalized with the return to glycolysis as the main energy-gaining process. Strong antibiotic treatment probably led to a decrease in urinary hippurate levels in COVID-19 patients. The urinary hippurate did not fully recover one month after hospitalization, which can point to still insufficient intestinal microbionta colonization one month after acute COVID-19 disease. Further, the increase in urinary levels of hypoxanthine in COVID-19 patients, most prominent one week after hospitalization, could be due to a particular time delay in hypoxanthine clearance reflecting backwardly the pathological processes, such as hypoxic and inflammatory tissue damage in the organism and disease severity. The initial decrease in alanine levels in urine, similar to those found in blood plasma in previous studies, suggests a slowdown in the nitrogen shuttle into the liver. In other words, alterations in the metabolic rate of an organism. This normalized with the time of treatment. Through the changes of other metabolites, other processes occurring in the body could be suggested, such as the normalization of initially decreased urinary citrate levels. This can be linked with the stabilization of mitochondrial energy metabolism during the tendency to recover the energy metabolism from the utilization of ketone bodies back to glycolysis. Further, the lowered utilization of tyrosine may result in the underproduction of thyroid hormones, as well as tyrosine-derived neurotransmitters such as dopamine and norepinephrine. 

Discriminatory analyses showed the feasibility of successful discrimination of COVID-19 patients from healthy controls based on urinary metabolites levels on the first day when admitted to the hospital, with AUC = 1, the fourth to the seventh day with AUC = 0.989, and one month later with AUC = 0.90, where the metabolites citrate, hippurate, and pyruvate were marked as of the highest importance. The last result shows that the metabolomic changes persist one month after the acute phase, and the organism’s recovery is not fully achieved.

## Figures and Tables

**Figure 1 metabolites-13-00364-f001:**
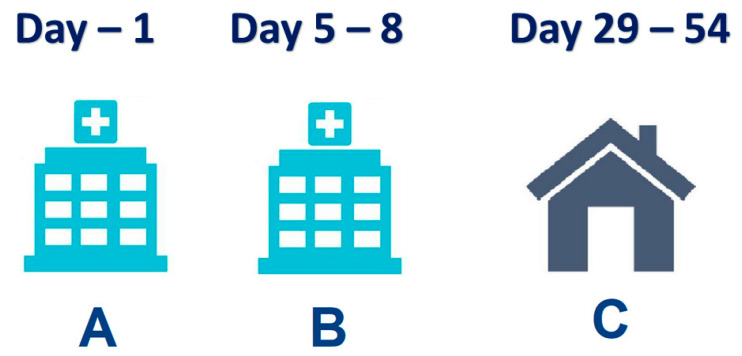
Timepoints of urine sampling from the COVID-19 patients.

**Figure 2 metabolites-13-00364-f002:**
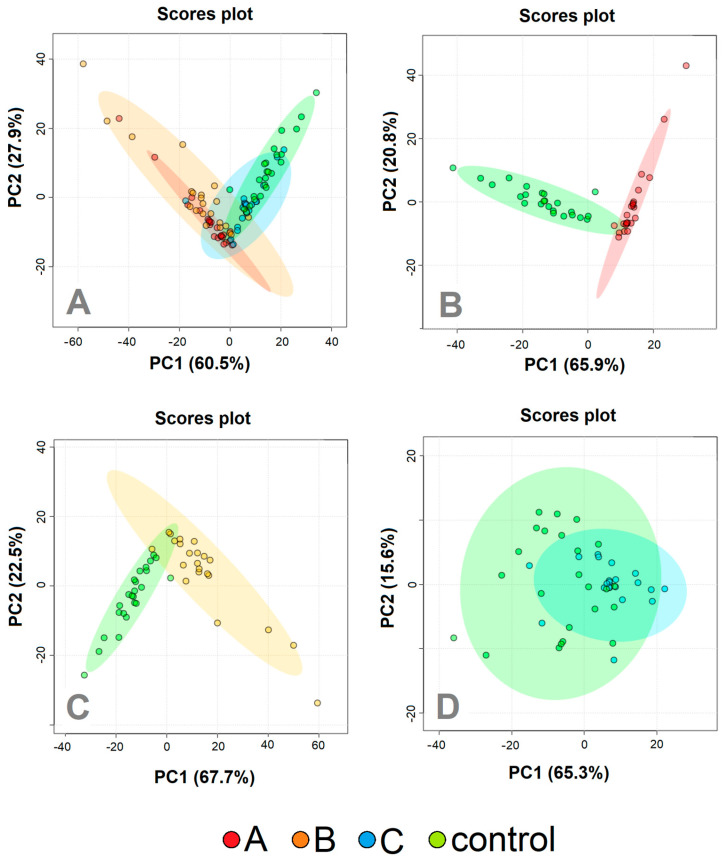
PCA analyzes patients with COVID–19 disease at three various sampling times (Day 1, red; Day 5−8, orange; Day 29−54, blue) and controls (green). (**A**) All three sampling times together, (**B**) Day 1 against controls; (**C**) Day 5–8 against controls, (**D**) day 29–54 against controls. The relative concentrations of metabolites in urine were used as input variables for the algorithm.

**Figure 3 metabolites-13-00364-f003:**
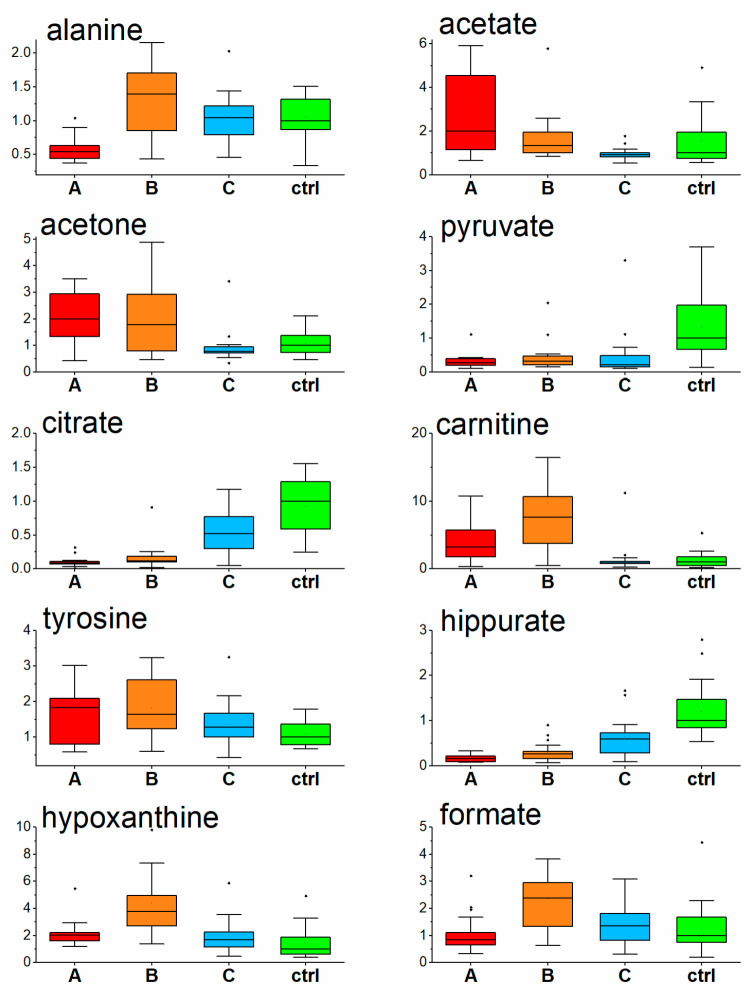
Relative levels of urine metabolites in COVID-19 patients at three consecutive timepoints and controls, data normalized to the level of creatinine, values related to the median of controls set to 1. (A-Day 1; B-Day 5–8; C-Day 29–54, ctrl-controls).

**Figure 4 metabolites-13-00364-f004:**
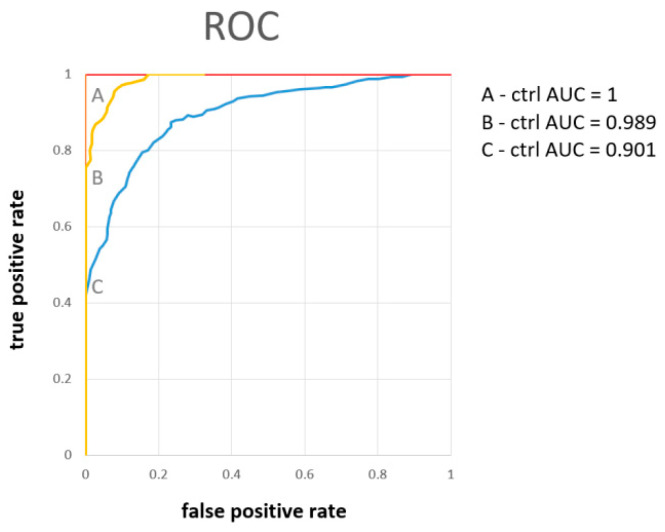
The ROC curve derived from the random forest discriminatory algorithm for binary systems patients on the first day of hospitalization (**A**), one week (**B**), and one month later (**C**), as input variables were used concentrations of urine metabolites related to the level of creatinine.

**Table 1 metabolites-13-00364-t001:** Characteristics of patients enrolled in the study.

	Median (Interquartile Range)
Age [years]	58 (21)
Sex: Female/Male	7/18
Weight [kg]	82.6 (26)
Height [cm]	171 (8)
BMI	29 (9)
Chronic liver disease	3
Chronic kidney disease	3
Ischemic cardiac disease	3
Diabetes Mellitus	3
Thyroidal disease	4
Rheumatic disease	0
Other relevant	NA

**Table 2 metabolites-13-00364-t002:** Standard biochemical and hematological results of patients at the sampling times, median (interquartile range), m indicates the number of missing samples.

	Samples A	Samples B	Samples C
Na^+^	133 (5)	140 (6)	139 (2)
K^+^	4.1 (0.6)	4.2 (0.7)	4.2 (0.4)
Cl^−^	101 (5)	104 (5)	104 (2)
Glucose	7 (1.4)	5.8 (29), 1 m	5.5 (1.1)
Creatinine	71 (29)	69 (23), 2 m	68 (24)
CRP	102.2 (124.8)	16.6 (38.1)	2.2 (4.4)
AST	0.88 (0.71)	0.93 (1.07), 7 m	0.575 (0.49)
ALT	0.805 (0.63)	1.465 (1.325),7 m	0.89 (1.09)
GMT	0.93 (2.08)	1.865 (3.605), 3 m	1.255 (0.46), 1 m
Bilirubin	10.7 (5.4)	11.1 (4.9), 8 m	9.5 (6.3)
Leukocytes	6.5 (2.9)	7.65 (3.5), 1 m	8.0 (3.1)
Hemoglobin [g/L]	143 (12)	136 (17), 1 m	144 (14)
Platelets count	177 (163)	355 (198), 1 m	259 (145)

**Table 3 metabolites-13-00364-t003:** Result from binary discrimination by random forest algorithm, the relative levels of urinary metabolites were used as the input variables.

	AUC	Number of Variables (Metabolites)	Metabolites by Importance	Average Accuracy Based on 100 Cross Validation	Oob Error
A-ctrl	1	2	hippurate, citrate	0.987	0
	1	5	hippurate, citrate, pyruvate, alanine, hypoxantine	0.987	0
B-ctrl	0.989	2	citrate, hippurate	0.931	0.09
	0.993	5	citrate, hippurate, carnitine, hypoxantine, pyruvate	0.936	0.09
C-ctrl	0.844	2	pyruvate, hippurate	0.784	0.32
	0.874	5	pyruvate, hippurate, citrate, glycine, hypoxantine	0.796	0.32
	0.901	all	pyruvate, hippurate, citrate, glycine, hypoxanthine, further metabolites were of comparable importance	0.775	0.28

**Table 4 metabolites-13-00364-t004:** The *p*-values from the statistical comparison of relative levels of urine metabolites (related to the abundance of urine creatinine) in patients on the first day of hospitalization (A), one week (B), and one month later (C) and controls (ctrl). For significant changes (*p*-value <0.0083 after Bonferroni correction), a percental difference was calculated using the medians.

	All	A-B	A-C	A-Ctrl	B-C	B-Ctrl	C-Ctrl
acetone	0.0012	0.65	0.00078 (152%)	0.011	0.0031 (121%)	0.035 (80%)	0.3
alanine	0.00033	0.00003 (−59%)	0.0026 (−47%)	0.001 (−46%)	0.17	0.15	0.96
acetate	0.002	0.33	0.00037 (140%)	0.016	0.0082 (62%)	0.15	0.21
pyruvate	1.7 × 10^−6^	0.28	0.75	0.0000034 (−74%)	0.45	0.00036 (−69%)	0.000016 (−79%)
citrate	3.58 × 10^−10^	0.25	0.00036 (−84%)	0.00000000057 (−91%)	0.013	0.00000041 (−88%)	0.019 (−49%)
carnitine	3.20 × 10^−7^	0.22	0.001 (282%)	0.00086 (233%)	0.0000058 (797%)	0.0000031 (646%)	0.87
tyrosine	0.0077	0.58	0.25	0.011	0.086	0.0015 (65%)	0.19
hippurate	1.90 × 10^−10^	0.22	0.00075 (−75%)	0.00000000031 (−85%)	0.024	0.00000022 (−75%)	0.0068 (−43%)
hypoxanthine	2.40 × 10^−5^	0.021	0.41	0.018	0.0017 (130%)	0.0000014 (282%)	0.13
formate	0.0028	0.00042 (−64%)	0.16	0.37	0.033	0.0034 (139%)	0.54

## Data Availability

All data, including raw and evaluated NMR spectra, are available on request: eva.baranovicova@uniba.sk. The data are not publicly available due to the nature of the research and sensitivity of information obtained from the research subjects.

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
