# Peer review of "Changes in the Urine Metabolomic Profile in Patients Recovering from Severe COVID-19"

_metabolites, 2023, doi:10.3390/metabo13030364_

Round 1

Reviewer 1 Report

The paper describes the urinary metabolomics in the patients with COVID19 and the development of biomarkers and discrimination models.

NMR was used as metabolome analytical method and accurate analysis was performed by it.

The urine have been sampled in three timing (A, B and C) and they are useful for the biomarkers development.

I have only a few comments.

1) I wonder what metabolites were used in model C development.

2) The equations for descrimination in A, B, and C might be stated. 

Author Response

Please, see the attached file with author´s response. Thank you very much.

Reviewer 2 Report

Rosolanka et al. attempted to investigate the changes in urine metabolites for severe COVID-19 patients who recovered. To achieve this, they leveraged the urine samples of 25 COVID-19 patients reported in one of their previous research. Urine samples across three time points over the course of COVID-19 infection are sampled: (1) within 24 hours of hospital admission, (2) 5 to 8 days after the initial hospitalization, and (3) 29 to 54 days after the collection of the first urine sample. Similarly, one single urine sample was collected for 24 healthy persons and used for the healthy control. The metabolome (i.e., the profile of metabolite concentrations) of urine samples was measured by NMR, and the metabolomes across different time points and groups were compared via either PCA or Random Forest. Those discriminatory analyses validated the feasibility of discriminating COVID-19 patients from healthy control based on urinary metabolomes. Finally, they examined the significance of each metabolite in the Random Forest model to discriminate and found that citrate, Hippurate, and pyruvate play the most important roles. I think this article is interesting for readers to know about the potential of urine metabolomes to discriminate COVID-19 patients from healthy controls. I am quite open to looking at a revised version if the authors could address some major and minor issues in a satisfactory fashion, which we describe in more detail below.

Major concerns:

1.     It is great that the authors demonstrated the power of urine metabolomes to discriminate COVID-19 patients from healthy controls. But why does this discriminatory power behave compared to other approaches such as using serum metabolite levels? They measured the serum metabolite levels are often used on line 57. Is it possible to compare their results to those that used the serum metabolite levels?

2.     I think a more detailed discussion over why and how the virus impacts the metabolism is lacking on lines 41-42. Please do a summary of previous papers that investigated the potential influence of viruses and cite those papers. For instance, does viral infection change host factors (Logan Miller et al., Viral Immunology 2022)? Or does viral infection change the gut microbiome (Shanlin Ke et al., Nature Communications 2022) then the gut microbiome changes the metabolism (Tong Wang et al., PloS Computational Biology 2019)?

3.     Many papers claimed to be able to predict metabolite levels using gut microbiomes. Please cite those papers to point out the potential connection between the gut microbiome and metabolome (Himel Mallick et al., Nature Communications 2019; Akshit Goyal et al., Nature Communications 2021; Derek Reiman et al., PloS Computational Biology 2021).

Minor comments:

1.     Show the full name for IQR.

2.     The legends in Figure 2 are too small to be visible.

Author Response

(The authors gave the same response as above.)

Reviewer 3 Report

The manuscript by Setti et al; "Changes in urine metabolomic profile in patients recovering 2 from severe COVID-19" is an important follow-up study after severe COVID-19 infection to identify metabolic markers persistent with the diseases and the extent of recovery. Such studies provide the after-effect of the diseases at a systemic level and help in understanding how to manage the long symptoms and potential health hazards of COVID-19 infection in people. The manuscript is scientifically sound and well-designed and the results are statistically well-analyzed and presented. The results are well analyzed and written with relevant possible causation and relevant references.  The manuscript could have been written more carefully avoiding avoidable spelling and syntax errors.  These are some suggestions and questions on the manuscript:

1. There are a lot of syntax errors all over the manuscript which at times makes it hard to understand the information. For example, lines 27-29, 43, 58-59, 93, 221-222, 277,297, 350.

2. The manuscript is heavily dependent on the previously published article by the authors (Ref. 14), if the samples were collected from the same group of patients and at the same time, analysis was done in a similar fashion, why was the study is split and presented in two different articles?

3. In line 335, the authors write human mammal cells, the context needs to be clarified or the sentence should be corrected.

4. Figure 4, hypoxanthine is written "hipoxanthine".

Author Response

(The authors gave the same response as above.)

Round 2

Reviewer 2 Report

The authors addressed my concerns satisfactorily. I have no further questions.